# An Indoor DFEC Ranging Method for Homologous Base Station Based on GPS L1 and BeiDou B1 Signals

**DOI:** 10.3390/s20082225

**Published:** 2020-04-15

**Authors:** Heng Zhang, Shuguo Pan, Chuanzhen Sheng, Xingli Gan, Baoguo Yu, Lu Huang, Yaning Li

**Affiliations:** 1School of Instrument Science and Engineering, Southeast University, Nanjing 210096, China; 13582161539@163.com; 2State Key Laboratory of Satellite Navigation System and Equipment Technology, Shijiazhuang 050081, China; shengchuanzhen@163.com (C.S.); ganxingli@163.com (X.G.); yubg@sina.cn (B.Y.); 18642720668@163.com (L.H.); 15631149037@163.com (Y.L.); 3The 54th Research Institute of China Electronics Technology Group Corporation, Shijiazhuang 050081, China

**Keywords:** indoor positioning, dual-frequency, homologous base station, ranging

## Abstract

High-precision navigation and positioning technology for indoor areas has become one of the research hotspots in the current navigation field. However, due to the complexity of the indoor environment, this technology direction is also one of the research difficulties. At present, our common indoor positioning methods are WIFI, Bluetooth, LED, ultrasound and pseudo satellite. However, due to the problem of inaccurate direct or indirect ranging, the positioning accuracy is usually affected, which makes the final application difficult to achieve. In order to avoid the ranging limitations of the existing methods, a new dual-frequency entanglement constraint (DFEC) ranging method based on homologous base station is proposed in this paper. The relationship between the homologous characteristics of dual-frequency signals and the phase relationship within the cycle is used to estimate the current carrier phase adjustment the true value of the cycle count is used to get rid of the constraints of the ranging conditions and improve the ranging accuracy. In order to verify the feasibility of this method, the wired environment test and the typical characteristic points of wireless environment are tested and analyzed respectively. The analysis results show that in the wired environment, the transmitting base station and the receiving terminal will introduce a ranging error of one wavelength; in the wireless environment, due to the influence of spatial noise and multipath, the error of the estimation of the whole cycles of the ranging value increases significantly. And this phenomenon is most obvious especially in the region where the signal is shaded, but the error estimate that satisfies ± 1 wavelength still accounts for 90%. Based on this, we conduct multiple observation data collection at five typical feature points, and used existing MATLAB positioning algorithms to conduct positioning error tests. The analysis found that under this error condition, the positioning accuracy was about 0.6 m, and 93% of the points met the 1-m positioning accuracy.

## 1. Introduction

Indoor navigation and positioning technology can be understood as the use of various technical means to achieve the positioning and tracking of people and objects in indoor spaces. With the development of the Internet of Things technology, there are more and more positioning needs of people and objects in economic behavior, personal activities, and military applications. In the outdoor environment, the global satellite navigation system is a universal and mature positioning method that allows people to achieve navigation and positioning outdoors. However, with the increasing indoor space today, the continuous construction of various factory workshops, large shopping malls, office buildings, and subway stations, indoor-based navigation and positioning is essential for the safety and monitoring of people and objects. The demand for location services in an indoor environment has become increasingly significant. People’s location service needs in indoor environments have become increasingly significant. According to the country’s future comprehensive PNT system construction needs, indoor navigation will be one of the three major national development directions. Scholars at home and abroad have conducted a lot of exploration and research. The previous positioning methods are roughly divided into three categories: sensor positioning based on recursive navigation, positioning based on feature point matching, and positioning based on base station signal ranging. The three methods have been developed in recent years, but there are still constraints. Here, this article only discusses the way of base station ranging

In the environment of indoor complex spatial characteristics, the biggest problem facing the current base station ranging method is the problem of inaccurate ranging. Due to the complex indoor spatial environment and serious signal multipath effects, the ranging values obtained indoors are usually difficult to meet the positioning needs.

In the current positioning technology means, there are two methods of direct ranging and indirect ranging. For example, positioning methods based on signal strength and feature matching, we can consider that indirect ranging methods are used for location estimation. Ahmed, AU uses the combination of WIFI and Bluetooth data for location research [1], Ren, J Sharp, I. and Yang, C. explores the mechanism of indoor location through algorithm combination based on signal strength [2,3,4]. Vazhenin, N.A. used mobile base stations to perform positioning tests by modifying existing WIFI signal characteristics [5]. Wang, B. and Liu, Y. Improve the WIFI positioning accuracy from the perspective of proximity weighted matching and distance constraint by constructing a fingerprint database [6,7]. Zhou, M. In the research process, the recursive principle of PDR was used to improve the indoor positioning service through the combination of recursion and WIFI correction [8]. Guo, G. Using the principles of WIFI RTT and RSS ranging, combined with the hyperbolic positioning algorithm, to carry out related positioning research on mobile phones [9]. However, at present, the positioning accuracy based on WIFI is still poor, and the positioning accuracy is usually 5–10 m. Visible light, which uses the intensity of light signal arrival or the flicker frequency of the light signal to achieve indoor positioning, Mocan, A. and Pelts, D.’s technical team analyzed the factors affecting the positioning accuracy from multiple angles [10,11]. The team of Pelts, D. analyzed the influence of this phenomenon on positioning based on the instability of visible light power [12]. Kim, D.R. used the RF Carrier Allocation Technique to construct a visible light system that integrates communication and navigation [13]. Wang, T.Q used visible angle to achieve indoor positioning [14,15]. This method has good positioning accuracy in a certain area, but requires a specific lamp design, so there is difficulty in promotion. The principle of pseudolite positioning is the same as that of GNSS, but due to the complexity of the indoor environment, there is also a problem that the range value jumps and the range measurement is not accurate. In the research process of Wuhan University, by constructing an indoor RTK method, using the method of UKF and known initial points to estimate the ambiguity of whole cycles and improve the ranging accuracy to achieve positioning, but in actual tests, this method is only suitable for simple indoors Environment, the difference correction value in complex environment is generally difficult to assist the ambiguity analysis [16,17,18,19], the other is our unit, the pseudolite system designed by our unit is currently the leading domestic. Our unit is the first country to introduce BeiDou pseudo-satellite technology indoors, and has achieved good results. By constructing an array-type pseudo-satellite network indoors, utilizing the characteristics of the pseudo-satellites and the network-derived network, and controlling the consistency of multiple signals through PLL control. However, the system still has the problem of inaccurate ranging indoors. For this reason, we have designed two positioning algorithms, an innovative fingerprint location algorithm [20] and the doppler positioning algorithms with known initial points [21]. These methods can currently achieve indoor positioning services at the centimeter level by avoiding real-time raw ranging value. However, from an application-oriented perspective, we do not expect the existence of initial values. When we enter a place to navigate, letting customers find the initial point before providing services is not conducive to technology promotion to a certain extent. The most desired state should be the state of entering or navigating. Therefore, the accuracy of ranging is again concerned.

Here, we introduce the principle of ranging. We can understand it as a problem similar to the pursuit of multiple wavelengths. When the first frequency signal propagates a wavelength length, the second frequency signal propagates the sum of the wavelength length plus the length of the wavelength difference between two signals. When the first frequency signal propagates N wavelength lengths, the second frequency signal propagates N wavelength lengths plus N wavelength difference between the two signals, so we can infer how many full wavelength lengths the two signals currently propagate based on the phase distance difference within a cycle between the two signals. The sum is the propagation distance of the current position signal at the current moment. The ranging diagram is shown in Figure 1.

The remainder of this paper is organized as follows: Section 2 introduces the working principle of the dual-frequency multi-channel signal base station and describes the ranging process in detail. At the same time, the phase accumulation cross-cycle problem and the frequency selection principle used in the ranging process are analyzed. Section 3 carried out a series of tests on the feasibility of the DFEC ranging method from the transmitting base station side, receiver side, and influence of noise and multipath in the wireless environment, and obtained the relevant test results. Section 4 Combined with the existing test results in Section 3 above, we carried out related positioning experiments at the pseudo-satellite artificial intelligence test site, and gave the positioning error analysis under this ranging method. Finally, on the basis of the existing results, we summarize the technical advantages of the current method and give the follow-up research directions.

## 2. DFEC Ranging Principle 

### 2.1. Overview

A schematic of the multi-channel dual-frequency signal transmission base station is given in Figure 2. As shown in the figure. First, the time-frequency control module completes the local clock in real time, and outputs stable 1PPS and accurate reference clock frequency. The reference clock obtains the required dual-frequency signal reference frequency through the clock frequency division module. Under the control of 1PPS, each frequency baseband signal modulation is completed in the baseband signal generation module, and finally the RF signal combining output is completed by the up-conversion module. Since the signals of different frequencies of multiple channels are controlled by the same clock frequency, Since the carrier waves transmitted from the antennas are all generated by a single phase-locked loop (PLL), their wavelength and frequency are the same, so all signals have the same time-frequency characteristics. At the same time, for each frequency signal transmitted on the same channel, because the hardware transmission path of each signal is the same, we can consider that the equipment delay between the signal of the same channel is the same, so the measurement distance error is known.

### 2.2. Proposed Method

In the previous section, we briefly introduced the principle of the indoor homogeneous base station system used. In this section, we will introduce the DFEC ranging process in detail.

Normally, a receiver outputs a carrier phase as an observable, which is an integrated value of the number of whole cycles and the fraction of a cycle of a beat wave that arises between the received and receiver-generated carrier waves. If the carrier phase of the k-th frequency signal is expressed as $\phi_{k}$ (whose unit is cycles), it is modeled as
(1)$$\phi_{k}=\lambda^{-1}\left\|r_{t}^{k}-r_{u}\right\|+\lambda^{-1}c(\delta t-\delta T)-N^{k}+\varepsilon_{\phi }^{k}$$
where λ is the wavelength of the k-th frequency signal, $r_{t}^{k}$ is the position of the pseudolite antenna k (which is known), $r_{u}$ is the receiver position to be determined, c is the speed of light, δt is the clock bias of the receiver, δT is the clock bias of the homogeneous transmitting base station, $N^{k}$ is the integer ambiguity (which is an integer value that consists of the number of wave fronts existing between the homogeneous transmitting base station and receiver antenna and an integrated value of cycles of the beat wave mentioned above), and $\varepsilon_{\phi }^{k}$ is the observation error of the carrier phase. This kind of equation modeling the observable by using the geometric relation between the receiver and the homogeneous transmitting base station is called “observation equation”. Since the signals transmitted from the antenna are synchronized, combining the carrier phase observation measurement Equation (1), we can get the observation equations of the GPS L1 and BeiDou B1 signals from the same antenna as:(2)$$\{\begin{array}{c} \phi_{L_{1}}=\lambda_{L_{1}}{}^{-1}\left\|r_{t}^{1}-r_{u}\right\|+\lambda_{L_{1}}{}^{-1}c(\delta t-\delta T)-N_{L_{1}}+\varepsilon_{\phi_{L_{1}}}^{}\\ \phi_{B_{1}}=\lambda_{B_{1}}{}^{-1}\left\|r_{t}^{2}-r_{u}\right\|+\lambda_{B_{1}}{}^{-1}c(\delta t-\delta T)-N_{B_{1}}+\varepsilon_{\phi_{L_{1}}}^{} \end{array}$$
where $\phi_{L_{1}}$ is the carrier phase of the GPS L1 signal, $\phi_{B_{1}}$ is the carrier phase of the BeiDou B1 signal, the unit of the carrier phase observation equation above is the cycle. $\lambda_{L_{1}}$ is the wavelength of the GPS L1 signal, $\lambda_{B_{1}}$ is the wavelength of the BeiDou B1 signal, $N_{L_{1}}$ is the integer ambiguity of the GPS L1 signal, $N_{B_{1}}$ is the integer ambiguity of the BeiDou B1 signal, $\varepsilon_{\phi_{L_{1}}}^{}$ and $\varepsilon_{\phi_{L_{1}}}^{}$ are the noise errors of the two frequency signals.

The system of equations is converted to the distance equation program. Multiply both sides of the equation by the corresponding wavelengths as
(3)$$\{\begin{array}{c} \lambda_{L_{1}}\phi_{L_{1}}=\left\|r_{t}^{1}-r_{u}\right\|+c(\delta t-\delta T)-\lambda_{L_{1}}N_{L_{1}}+\lambda_{L_{1}}\varepsilon_{\phi_{L_{1}}}^{}\\ \lambda_{B_{1}}\phi_{B_{1}}=\left\|r_{t}^{2}-r_{u}\right\|+c(\delta t-\delta T)-\lambda_{B_{1}}N_{B_{1}}+\lambda_{B_{1}}\varepsilon_{\phi_{L_{1}}}^{} \end{array}$$

Let $\Phi_{m}=\lambda_{m}\phi_{m}$, then the above equation can be expressed as
(4)$$\{\begin{array}{c} \Phi_{L_{1}}=\left\|r_{t}^{1}-r_{u}\right\|+c(\delta t-\delta T)-\lambda_{L_{1}}N_{L_{1}}+\xi_{\phi_{L_{1}}}^{}\\ \Phi_{B_{1}}=\left\|r_{t}^{2}-r_{u}\right\|+c(\delta t-\delta T)-\lambda_{B_{1}}N_{B_{1}}+\xi_{\phi_{L_{1}}}^{} \end{array}$$
where $\xi_{\phi_{L_{1}}}^{}=\lambda_{L_{1}}\varepsilon_{\phi_{L_{1}}}^{}$, $\xi_{\phi_{B_{1}}}^{}=\lambda_{B_{1}}\varepsilon_{\phi_{L_{1}}}^{}$.

As above, we obtain the distance equations based on the carrier of the two frequency signals. Since each signal is generated by the same PLL loop control, it can be obtained that the signal transmission clock difference is the same, and the receiver clock at the receiving end is also the same. It can be found that by making the difference of the distance equations of the signals of two frequencies, we can eliminate the influence of the clock difference. Here we only give the difference equation when the frequency L1 signal is used as the reference. From this we get the following formula.
(5)$$\Phi_{B_{1}}-\Phi_{L_{1}}=\left\|r_{t}^{2}-r_{u}\right\|-\left\|r_{t}^{1}-r_{u}\right\|-\lambda_{B_{1}}N_{B_{1}}+\lambda_{L_{1}}N_{L_{1}}+\xi_{\phi_{B_{1}{,}_{L_{1}},}}^{}$$

From the above formula, we can find that the distance difference between the signals becomes independent of time, and only related to the real distance of the signal and the ambiguity of the whole cycles. As mentioned above, we can use the principle of phase chase. The intra-cycle phase is calculated to calculate the true whole-circle distance value. Therefore, the above equation can also be expressed by the sum of the whole-phase phase distance value and the phase distance value within a period, thereby avoiding the problem of ambiguity of the whole cycles. Similarly, assuming that we know the initial phase of two frequency signals, we re-express the formula 4 as
(6)$$\{\begin{array}{c} \Phi_{L_{1}}=\lambda_{L_{1}}*N+\Delta \phi_{L1}-\Delta \phi_{i,B1}+\xi_{\phi_{L_{1}}}\\ \Phi_{B_{1}}=\lambda_{B_{1}}*N+\Delta \phi_{B_{1}}-\Delta \phi_{i,B_{1}}+\xi_{\phi_{B_{1}}} \end{array}$$

Wherein, N is the whole cycles, $\Delta \phi_{L_{1}}$ and $\Delta \phi_{B_{1}}$ are the phase value within the cycle of the two frequency signals and $\Delta \phi_{i,L_{1}}$, $\Delta \phi_{i,B_{1}}$ are the phase value within the initial cycle of the two frequency signals. The definition of $\xi_{\phi_{L_{1}}}^{}$ and $\xi_{\phi_{B_{1}}}^{}$ is the same as formula (4).

Here, we only give the phase difference equations based on the frequency of GPS L1. The specific expression is as follows.
(7)$$\{\Phi_{B_{1}}-\Phi_{L_{1}}=(\lambda_{B_{1}}-\lambda_{L_{1}})*N+(\Delta \phi_{B_{1}}-\Delta \phi_{L_{1}})-(\Delta \phi_{i,B_{1}}-\Delta \phi_{i,L_{1}})+\xi_{\phi_{B_{1},L1}}^{}$$

Since all signals are sent from a unified antenna, the difference on the left side of Equation (7) can be 0, and the value of the whole cycles N can be obtained by the above formula.
(8)$$N=\frac{(\Delta \phi_{B_{1}}-\Delta \phi_{L_{1}})-(\Delta \phi_{i,B_{1}}-\Delta \phi_{i,L_{1}})+\xi_{\phi_{B_{1},L_{1}}}^{}}{\lambda_{L_{1}}-\lambda_{B_{1}}}$$

Combining the contents of Section 2.2 and Section 2.3, the detailed ranging process is given at the end of Section 2, the specific process is shown in Algorithm 1.
**Algorithm 1.**1:  **Initialization Parameters:**   Signal transmitting base station coordinates;               Ground calibration point coordinates;               Ground test point coordinates;  **Initial phase calibration:**
$\Delta \phi_{i,L_{1}}$
**,**
$\Delta \phi_{i,B_{1}}$
2:  **while**
ϕ not empty **do**3:   Group by frequency characteristics D←GroupbyFre();4:   Cycle count to distance conversion Φ←λϕ;5:   Get the distance difference between frequencies diff(Φ);6:   Phase accumulation span detection:**if**ΔΦ<diff(λ)    (n,index)=cycle_n_modi(cn0,doppler,position,Φ,indx);   **else**    n=0;   **end**7:    Calculate the whole cycles N:$N=calN(\Delta \varphi ,\Delta \varphi_{i},\lambda )$8:    Calculate the distance ρ
    $\rho =calP(\Delta \varphi ,\Delta \varphi_{i},\lambda ,n,index)$;9:  **end while**


### 2.3. Cumulative Over-The-Cycle Detection

When we use this ranging method, we must pay attention to one situation. When the phase difference value is accumulated over one cycle, the accumulated phase difference value may be larger than the wavelength of a signal having a higher frequency at a certain point. In this case, the higher frequency signal will appear distance compensation value for several cycles. At this time, we can combine information such as the signal-to-noise ratio, doppler, and the position of the previous moment to estimate the specific value of the compensation value added at the current position.

### 2.4. Frequency Selection Analysis

The DFEC method is of great significance in ranging, but not any combination of frequency values can meet the application requirements of this ranging method. Generally, we need to meet the following requirements when selecting a frequency:

The first point is that the frequency we choose must meet the requirements of the Radio Regulatory Commission and be compatible with the frequency signals used today;

Secondly, the frequency ratio relationship between the two frequency values we choose must satisfy more than 0.5 and less than 2. In this way, during the calculation of the cycle, the problem of ambiguous estimation of the cycle does not occur.

Third, after we choose the frequency, the resulting wavelength difference must be greater than the distance difference introduced by the phase error, and the ranging error reflected in the range of the phase error can meet the future sub-meter positioning requirements.

Fourth, we recommend that when the frequency is selected for distance estimation in this way, each time a higher frequency signal is used to compensate the cycle, the ranging value between two consecutive cycles will be several meters or more. In this way, when the phase cumulative value within the cycle spans the cycle, we can quickly estimate what the value is based on the carrier-to-noise ratio, doppler, and the position of the previous moment.

## 3. Implementations and Evaluation

In this section, we will use the existing test environment to verify the feasibility of the ranging principle. By using a dual-frequency pseudo-satellite system to broadcast the frequency of B1 (1575.42 Mhz) and L1 (1561.098 Mhz) pseudo-satellite signals indoors, use mature commercial receiver ublox to carry out indoor signals carrier phase data collection, and use the obtained carrier phase value to carry out ranging experiments. The whole experimental process is mainly considered from three aspects: first, whether each switch on and off has an impact on the estimation of whole cycles; second, whether the internal error of the receiver affects the estimation of the whole cycles after each cold start; if the above two points have no impact, then we need to further verify how much the environmental interference has affected the whole cycles estimation. After completing these three steps of verification, we can determine whether this method can actually be used for ranging indoors, so the entire experimental process involves two parts. The first part is to connect the pseudo satellite base station to the ublox receiver by wire, and using the u-center receiver to receive and output the observation data. Then use MATLAB program for analysis. Here, in order to complete the verification quickly, the existing analysis algorithm is used, and it is judged by analyzing the whole cycles estimate of the distance difference between the channels. If the value is consistent each time, it can be considered that the ranging method is feasible. The same analysis algorithm is also used in subsequent wireless environment data analysis. The second, by using the existing test environment, multiple typical feature points are selected in this environment, and then the feasibility and fitness of the method at different feature points are analyzed. The specific settings are described in the following sections.

### 3.1. Wired Connection Test

The connection of the wired test environment is shown in Figure 3. The channels of the transmitter are connected by RF cables of the same length and are combined to the receiver through the combiner signal. The right part of the figure below is the status of the information output by the ublox receiver. The information output by the data window includes pseudolite number, pseudorange, carrier phase, doppler and carrier-to-noise ratio and so on. The part of the bar graph is the carrier-to-noise ratio of the receiver output.

A. Test Results of Switching Pseudolite Base Station

In this section, we mainly analyze the influence of the switch on the signal. Each time the machine is turned on or off, the time-frequency characteristics of the transmitting base station, the signal characteristics of the baseband module and the radio-frequency module are more or less affected. Therefore, it is necessary to analyze the influence of the signal error introduced on the ranging estimation through the switch. Here, we give the data processing results after switching the pseudolite base station three times, including the intra-cycle phase difference between L1 frequency channels, the intra-cycle phase difference between B1 frequency channels, and the whole-cycle estimated distance between channels. It can be seen from the following Figure 4 that the phase difference between the channels of the L1 output after the power-on and turn-off is basically the same, and there is a phase error of ± 0.005 ~ ± 0.01 cycles. The phase difference between the channels of the B1 in the Figure 5 remains the same every time the machine is turned on and off, and there is a phase error of ± 0.01 ~ ± 0.02 cycles. From this we get the estimated value of the whole cycles of the distance difference in Figure 6. It can be seen from the figure that the count of whole cycles has an uncertainty of one cycle, that is, one cycle of ranging errors may be introduced.

B. Cold Start Receiving Terminal Test

In this section, we will mainly analyze the impact on the estimated count of the whole cycles from the receiving end. With each cold start, the receiver will reacquire the signal on the basis of no prior information, so each time the signal is allocated to the channel after the replenishment, it will also be randomly allocated. This experiment can verify the influence of channel consistency on the the estimated count of the whole cycles. Of course, it also includes the influence of the measurement error introduced by the receiving terminal itself.

Here we give the data processing results after three cold starts. Figure 7 shows the phase difference between the channels at the L1 frequency and Figure 8 shows the phase difference between the channels at the B1 frequency. It can be seen from the figure that the phase difference between the two signals output after the three cold starts is basically the same, and the phase difference error of the cycle at the L1 frequency is ± 0.005 ~ ± 0.01cycle. The phase difference error of the cycle at the B1 frequency is ± 0.005 ~ ± 0.01cycle. On this basis, we get the the whole cycles estimate of the distance difference between the channels, as shown in Figure 9. The whole cycles estimated distance difference between the channels is basically the same, but there is one wavelength estimation error.

From the above analysis, we can find that, under the condition that the signal link is not affected by the outside world, the whole cycles estimate of the distance difference between the channels of the pseudo-satellite transmitting base station and the receiving terminal is the same, which together introduces one-cycle ranging error. In the following analysis, we will introduce the space environment impact and analyze the ranging estimation situation in the case of wireless transmission

### 3.2. Wireless Environment Test

A. Test Environment Introduction

In this section, we will carry out further tests in a wireless environment. The test scenarios are selected in our pseudo-satellite navigation test field, as shown in Figure 10. The upper left corner of the picture is our transmitting base station, the upper right corner is our multi-scenario analog antenna array, and the lower left corner is our test area. This area can simulate a variety of scenes such as viewing scenes, semi-occluded scenes and fully-occluded scenes. We carry out testing under a variety of conditions to verify requirements. The lower right corner is our test equipment. In order to avoid the introduction of human behavior jitter to evaluate the ranging results, we choose to use a small cart to evaluate the ranging effect of each characteristic test point.

Figure 11 is a floor plan of the test hall environment on the first floor. Here we selected five feature points, of which 21 and 23 are the two points in the viewing area, which are used to simulate the ranging effect in different positions in the viewing environment. Point 30 is the boundary area for viewing. In the existing tests, the positioning accuracy of this point is usually difficult to guarantee. As a feature point, we want to further evaluate whether the distance measurement method can solve the positioning problem at this point in the future. Point 1 is a semi-shielded area of the signal, but since the area is close to the stairs, the multipath phenomenon at this point is usually obvious. Point 26 is a fully shielded area, in which we can’t see the position of the transmitting antenna at all. Through the analysis of the five characteristic points, we hope to find out whether the current dual-frequency ranging method can improve the impact of cycle slip and multipath on ranging to some extent. At the same time, in order to further verify the generality of the algorithm, we divided the process into two rounds during the test. After completing the data collection of five points in the first round, the second round of data collection was performed. This can reduce the probability of receiving the same signal at a fixed point. See the following sections for specific test analysis.

B. Results of Wireless Environmental Testing

Figure 12 is an estimated value of the distance difference between channels obtained by analyzing the three-point data of the through-view area 21, 23, and 30 collected in the first set of tests according to the whole number of wavelengths. Figure 13 is an estimated value of the distance difference between channels obtained by analyzing the three-point data of the through-view area 21, 23, and 30 collected in the second set of tests according to the whole number of wavelengths. It can be found from the figure that the estimated value of the whole cycles obtained in the wireless case is significantly more frequent than that in the wired case. The two estimates of 21 points, 23 points, and 30 points each have a ranging error of ± 2 cycles.

Figure 14 and Figure 15 are the estimated values of the whole cycles of the channel distance difference in the two sets of tests of the signal half-shielded area 1 test point and the full-shielded area 26 test point. By comparison, it can be found that because the signal is affected by occlusion and signal multipath, the estimated value of the whole cycles is more fluctuating relative to the viewing environment, and the more severe the occlusion is, the more frequent the estimated value is. There is a ranging error of ± 3 cycles at 1 point, and a ranging error of plus or minus 5 cycles at 26 points.

In order to further analyze the whole cycles estimated characteristics of the method, we perform a probability statistical analysis on all the data, where Figure 16 is the probability statistical distribution map of 21, 23, and 30 points in the viewing area. Figure 17 shows the probability distribution of point 1 and point 26 in the masked area. The result show that in two environments, the viewing area and the obscured area, the probability of the range error of the channel-to-channel distance difference based on the whole-cycle count of the wavelength being within ± 1 cycle was still at least 90%.

## 4. Positioning Analysis

Through the above analysis, we can conclude that the maximum carrier phase difference fluctuation is within ± 0.01 cycles, and that the ranging error based on the entire wavelength count estimate of the ranging error meets a probability of 90% within ± 1 cycle. The worst ranging error of the distance difference between channels is about 4 cycles. In order to further verify the feasibility of this ranging method in subsequent positioning, we performed static positioning tests at the above five typical test points and compared them with the actual position parameters. The data output frequency was 5 Hz and the sampling time was 5 min. Figure 18 below shows the positioning error of a test point and a range that satisfies the positioning accuracy of 1 m. It can be seen from the figure that although there is a large fluctuation in the estimation of the whole cycles in the previous ranging stability test, it can still provide high-precision positioning by combining with the positioning algorithm. Based on this, we have carried out probability statistics on multiple positioning. Figure 19 shows that in the case of such ranging errors, the final positioning error is mainly about 0.6 m, and about 93% of the points meet the positioning accuracy of 1 m.

## 5. Conclusions

Aiming at the problem of inaccurate ranging by current indoor positioning methods, it is difficult to meet the problem of high-precision positioning. Based on the homologous base station, this paper proposes a DFEC ranging method. In this way, by using the phase constraint relationship between the dual-frequency signals, an equation for measuring the propagation distance using the phase within the cycle is constructed. This method has the following advantages: (1) It is not necessary to know the transmission time and reception time of the signal, and it avoids the dependence of ranging on time; (2) When a cycle slip occurs, carrier phase cycle slip is usually introduced, this method does not have the problem of the ambiguity of whole cycles; (3) Through testing in wired and wireless environments, it is found that in the semi-shielded and fully-shielded areas of the signal, although affected by the multipath signal, the ranging method can still meet the positioning accuracy of 93% probability. In future work, we will further use this ranging method to conduct comprehensive test verification on WIFI RTT, 5G and dual-frequency multi-system pseudo satellites. By analyzing the feasibility of various positioning methods, we will lay the foundation for further multi-source fusion basis. Thus, a ranging method using multiple means and multiple frequency phases to constrain is used to improve the accuracy and stability of indoor environment ranging and ensure the service demand of continuous reading high-precision positioning.

## Figures and Tables

**Figure 1 sensors-20-02225-f001:**
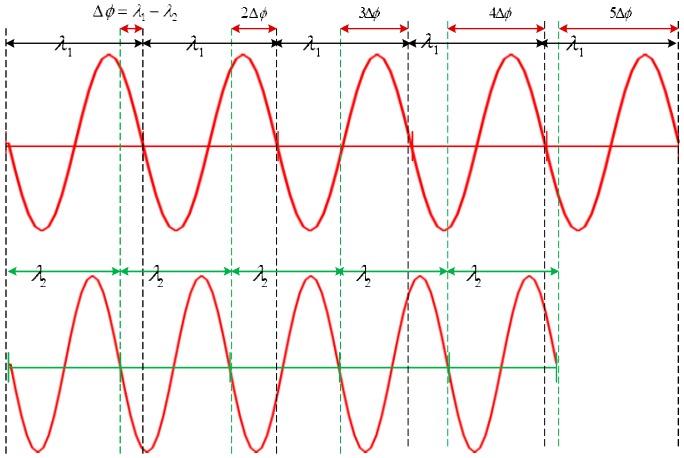
dual-frequency ranging principle.

**Figure 2 sensors-20-02225-f002:**
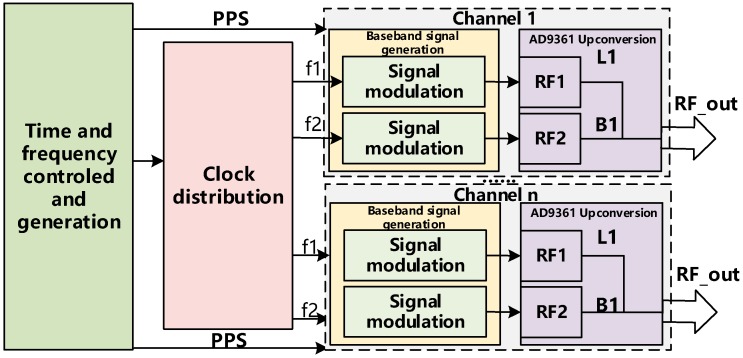
Transmission base station working principle.

**Figure 3 sensors-20-02225-f003:**
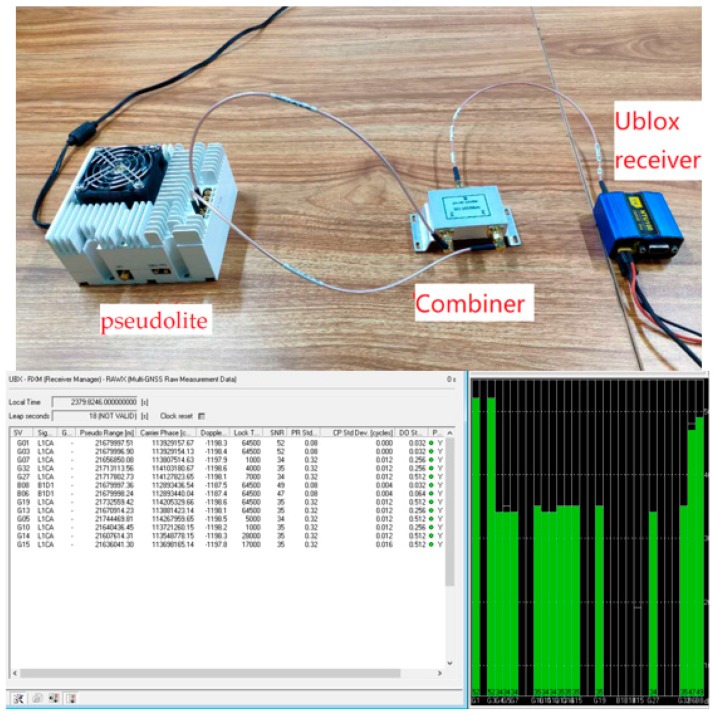
Wired environment test chart.

**Figure 4 sensors-20-02225-f004:**
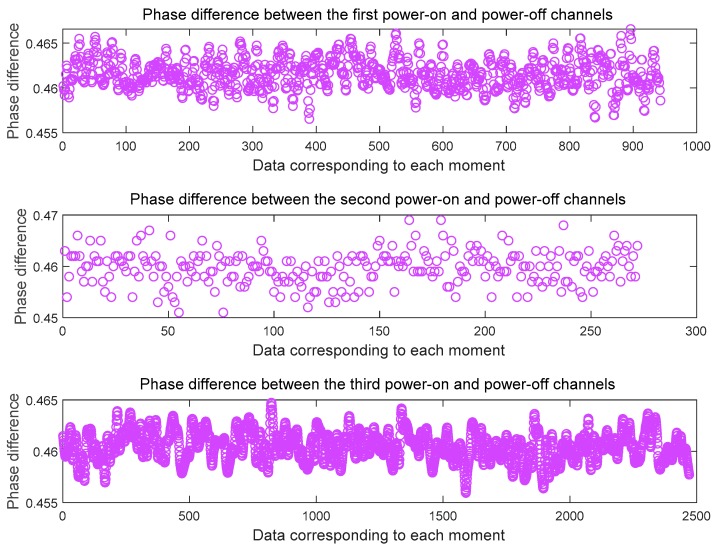
Test of influence of power on/off on L1 phase difference results between channels.

**Figure 5 sensors-20-02225-f005:**
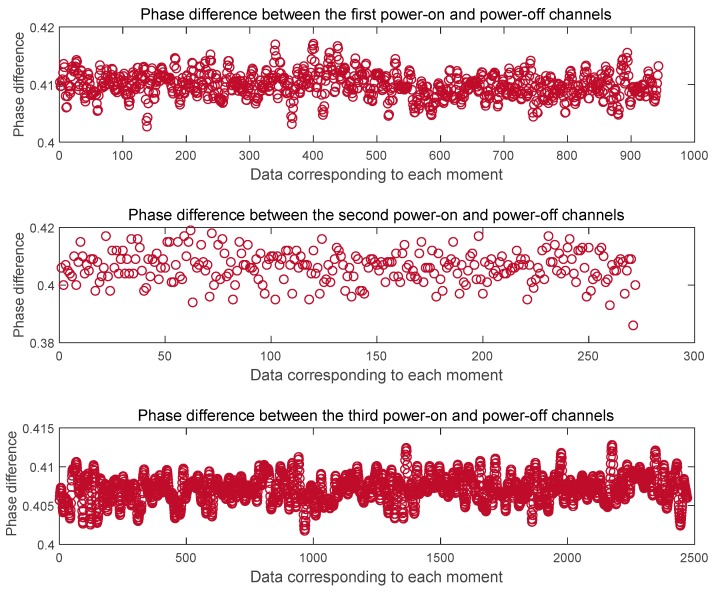
Test of influence of power on/off on B1 phase difference results between channels.

**Figure 6 sensors-20-02225-f006:**
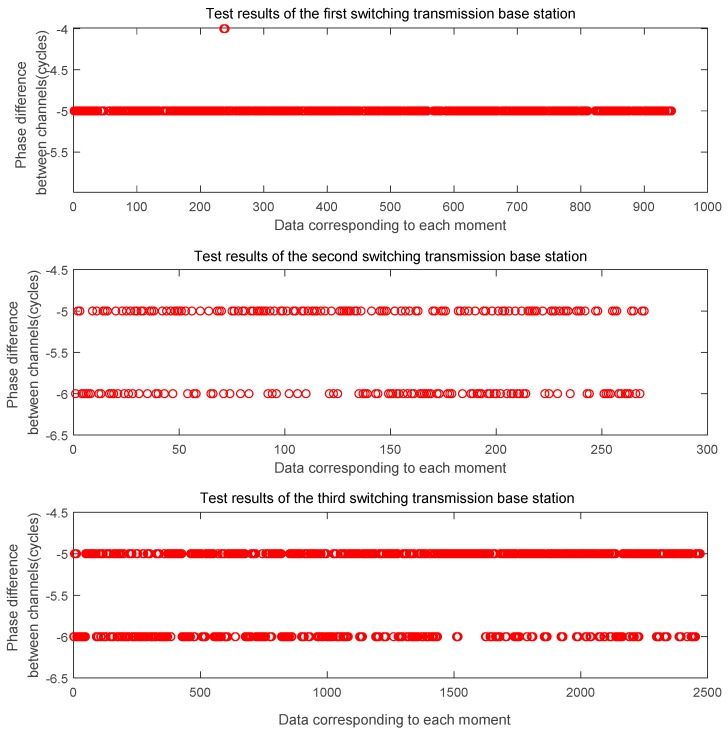
The distance based on the phase difference is estimated by the whole number of wavelengths.

**Figure 7 sensors-20-02225-f007:**
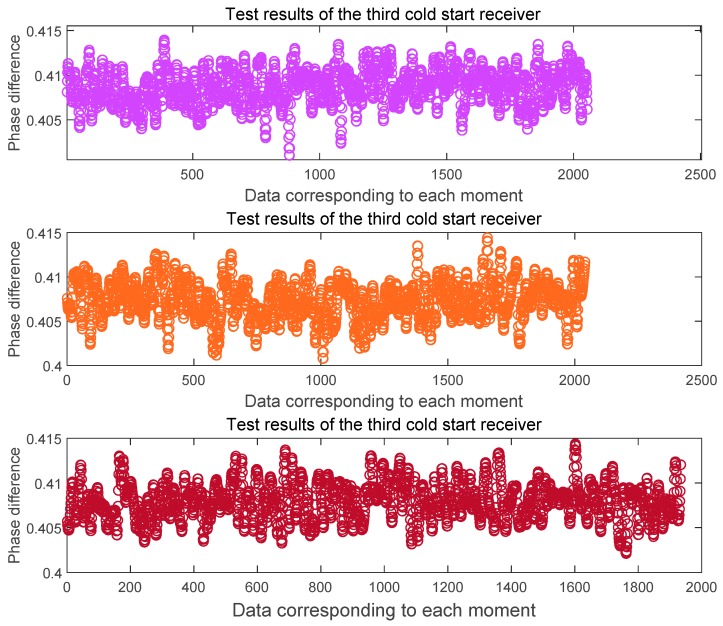
Phase difference between L1 channels after three cold starts of the receiver.

**Figure 8 sensors-20-02225-f008:**
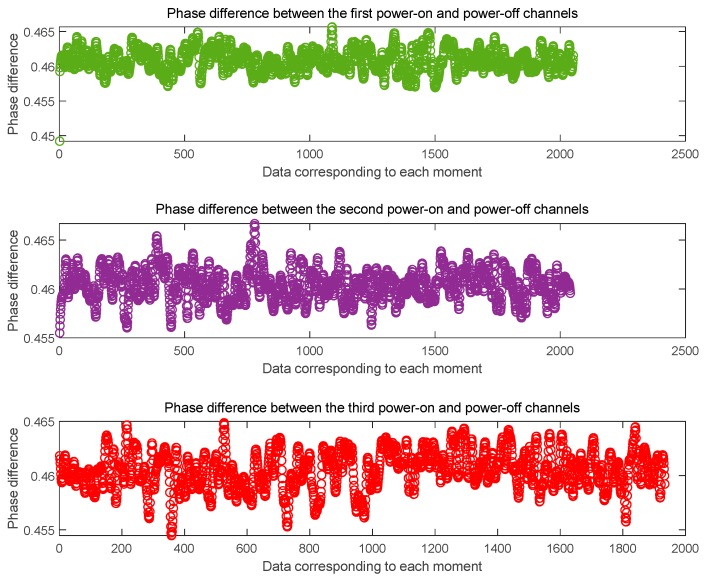
phase difference between B1 channels after three cold starts of the receiver.

**Figure 9 sensors-20-02225-f009:**
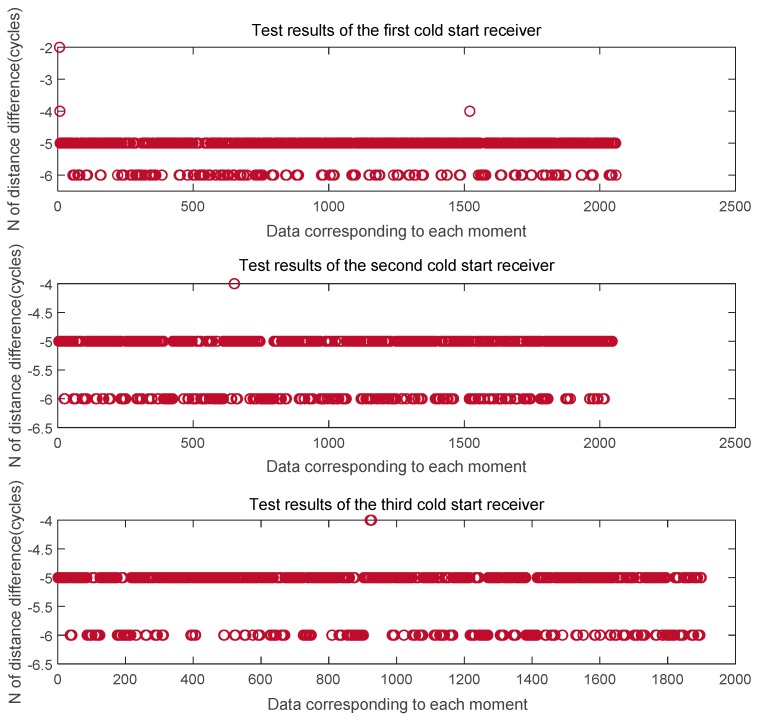
The distance based on the phase difference is estimated by the whole number of wavelengths.

**Figure 10 sensors-20-02225-f010:**
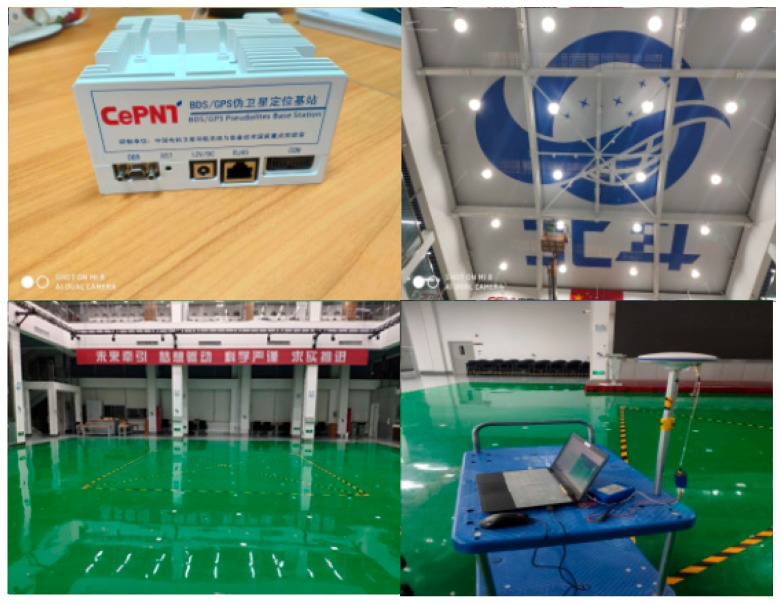
Wireless test environment.

**Figure 11 sensors-20-02225-f011:**
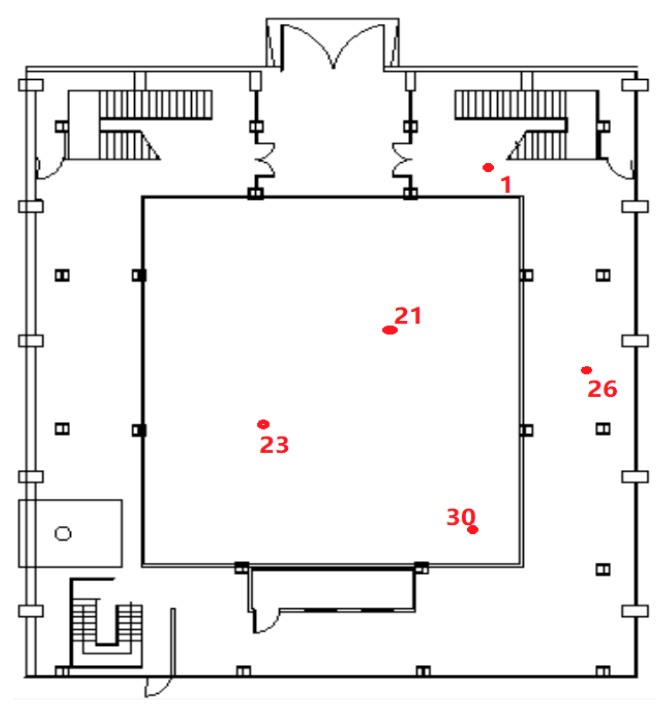
Distribution map of test feature points on the first floor.

**Figure 12 sensors-20-02225-f012:**
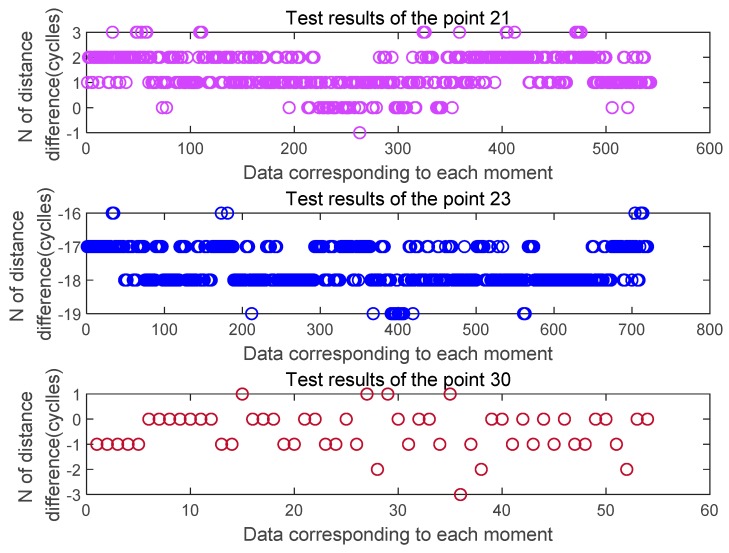
Estimated distance difference N at the three feature points in the first set of tests.

**Figure 13 sensors-20-02225-f013:**
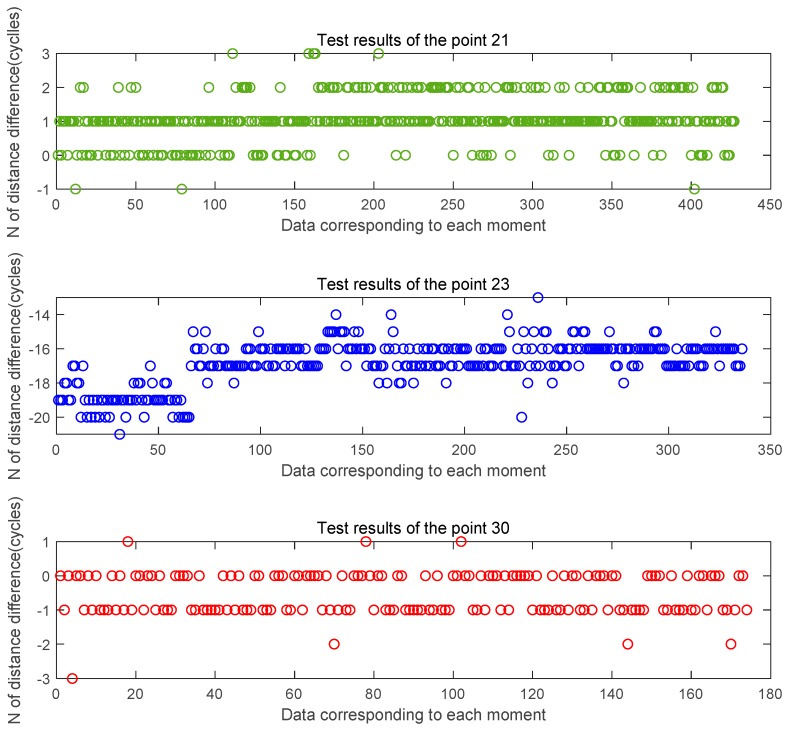
Estimated distance difference N at the three feature points in the second set of tests.

**Figure 14 sensors-20-02225-f014:**
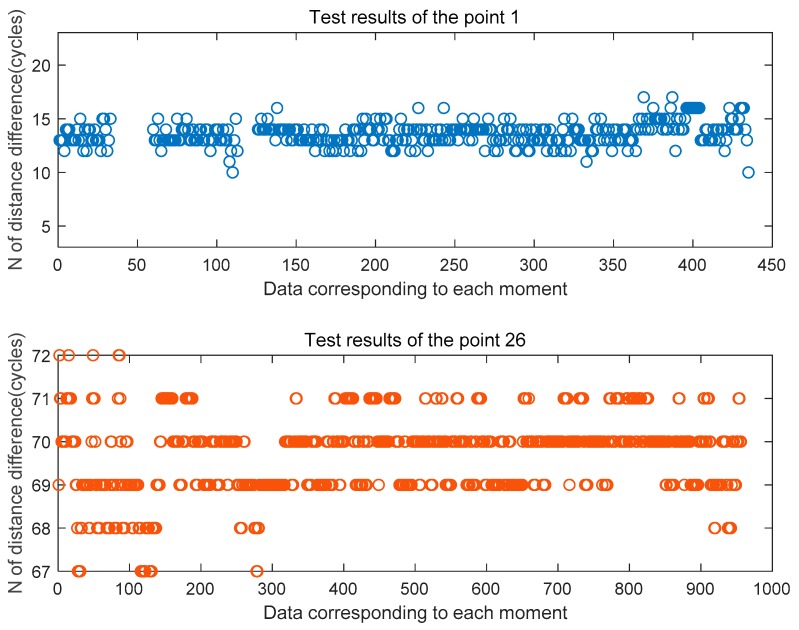
Estimated distance difference N at the three feature points in the first set of tests.

**Figure 15 sensors-20-02225-f015:**
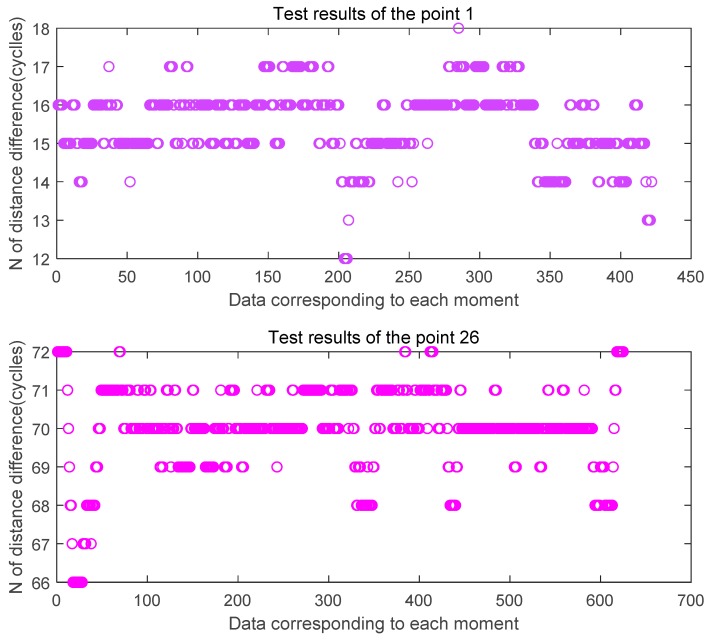
Estimated distance difference N at the three feature points in the second set of tests.

**Figure 16 sensors-20-02225-f016:**
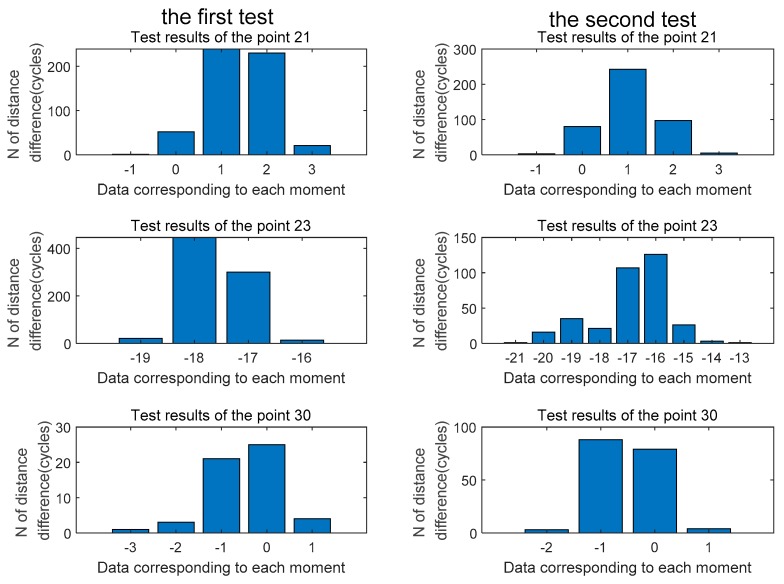
The probability statistical distribution map of 21, 23, and 30 points.

**Figure 17 sensors-20-02225-f017:**
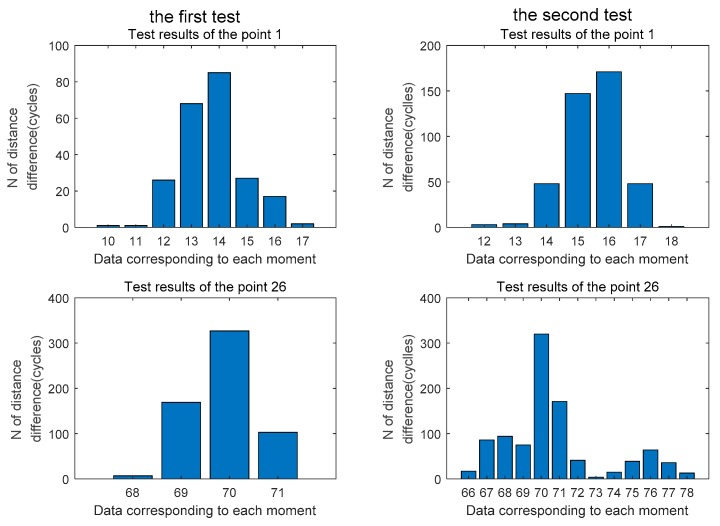
The probability statistical distribution map of 1and 26 points.

**Figure 18 sensors-20-02225-f018:**
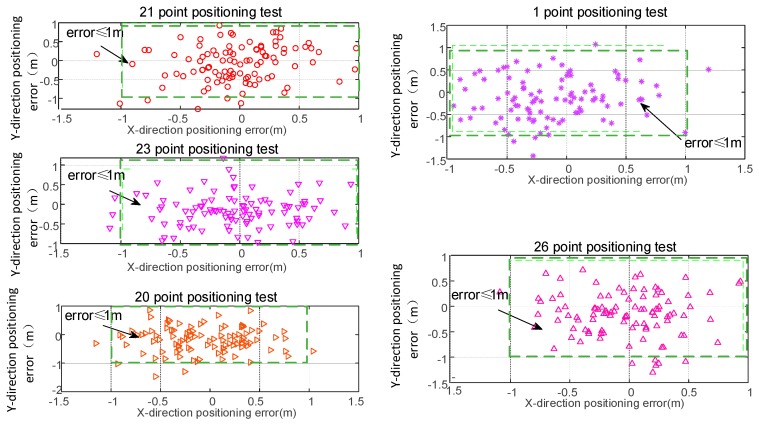
Positioning accuracy tests.

**Figure 19 sensors-20-02225-f019:**
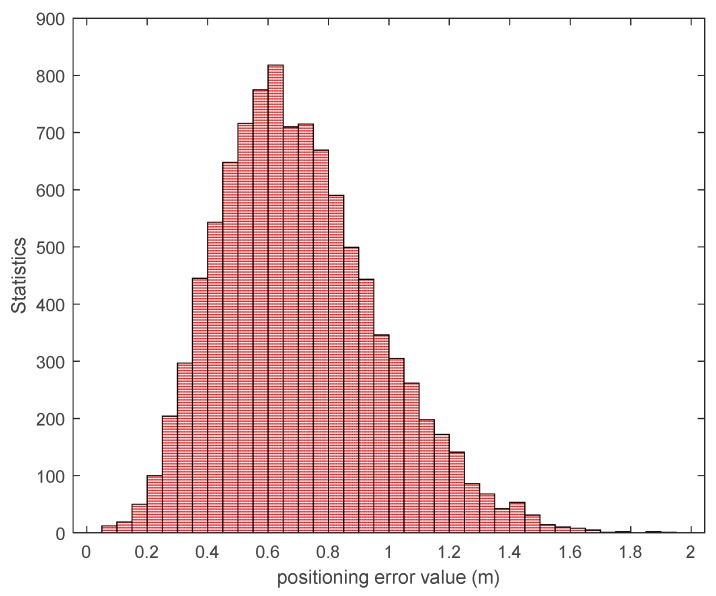
Probability statistics of positioning accuracy for multiple positioning tests.

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
