# Peer review of "An Indoor DFEC Ranging Method for Homologous Base Station Based on GPS L1 and BeiDou B1 Signals"

_sensors, 2020, doi:10.3390/s20082225_

Round 1

Reviewer 1 Report

Dear Authors,

I reviewed your article "An Indoor DFEC Ranging Method for Homologous Base Station based on GPS L1 and Compass B1 Signals" with interest.

I think the content of this paper is almost reasonable.
But I have some questions and comments, please respond to them.

----------------------------------------------------------------------
1) You call the Chinese positioning system "Compass", but I think it should be added to the sentence that the official name is BeiDou.

2) What does the "frequency point" on line 12 on page 3 mean?
What does the "frequency point signal" on line 8 mean?
If it has an unusual meaning, please define the meaning of the word before using it.

3) What does the "weekly distance" in line 11 on page 3 mean?
What does the "whole week" on the first line of page 15 mean?
I think this "week" does not have a general meaning. IF so, please define the meaning of this word before using it.

4) Some figures, such as Figure 1 and Figure 10, are too narrow at the top and bottom of the figure, between it and the text.
Please widen the gap.

5) I think the input part of f1 of Channel n in Figure 2 is wrong.
If so, correct it.

6) The third line from the bottom of page 5, Fomula 4, should be Fomula (4).

7) In the left half of Figure 3, the text in the image for illustration is too small, so make it larger.
Please explain the right half in the text or caption.

8) In Figure 4 ~ 5, Figure 7 ~ 9, Figure 12 ~ 18, the gaps above and below the three graphs are too narrow.
Please leave more space.

9) I think that you can calculate the true values of each of the experimental results shown in Figures 12 to 15.
But why not specify them?

10) What is the "first circle" at the end of the captions in Figures 12 and 14?
Similarly, what is the "second circle" at the end of the captions in Figures 13 and 15?
Please explain in the text.

11) The text in Figure 18 is too small to read. Please enlarge each graph.
----------------------------------------------------------------------
Sincerely yours,

Author Response

Hello teacher. Relevant opinions have been revised, Please see the attachment. Thank you.

Reviewer 2 Report

There are many phrases that are either grammatically incorrect or lack support. I provide some examples here:

1)  "limited by prerequisites" in your abstract. What do you mean ? Do you mean we simply have some limitations due to implementation design or something like that ? If not, please clarify.

2) "And the performance is most obvious in the area covered by the signal, but the error estimate that meets ± 1 wavelength still accounts for 90%." This is a very unclear phrase.

3) You mention repeatedly throughout the article "original data collections". What is an original data collection ? What is unoriginal data collection ? 

4) "Each factory workshop, large shopping mall, office." Is this supposed to be a complete sentence ? 

The derivations seem to be novel and this seems to work well with commercial receivers (ublox). Positioning accuracy is 0.6 meter, which is relatively decent for GPS receivers. A complexity analysis could be included, for future work maybe ? 

Given the above corrections, I recommend acceptance of this journal paper.

Author Response

(The authors gave the same response as above.)

Round 2

Reviewer 1 Report

Dear authors,

I have reviewed your revised paper.
Questions and ambiguities in the first version of the paper have been improved and clarified.
Therefore, I consider this paper is suitable for publication in the journal.

Sincerely Yours